# Performance Evaluation of a Full-Duplex Relaying-Enabled Satellite Sensor Network

**DOI:** 10.3390/s19245453

**Published:** 2019-12-11

**Authors:** Xigang Xia, Bo Yang, Zhiyu Liu, Kang An, Kefeng Guo

**Affiliations:** 1JiLin JLU Communication Design Institute Co. Ltd., Jilin 130012, China; persist_001@126.com (X.X.); liuzhiyujihua@163.com (Z.L.); 2National University of Defense Technology, Nanjing 210007, China; ankang89@nudt.edu.cn; 3Space Engineering University, Beijing 101416, China; guokefeng.cool@163.com

**Keywords:** satellite sensor network, full-duplex, residual loop interference, outage probability, ergodic capacity

## Abstract

This paper investigates the performance of a full-duplex (FD) relaying-enabled satellite sensor network under residual loop interference, where the satellite uplink and the downlink transmissions simultaneously occur over the same frequency band. Specifically, the closed-form expressions for the outage probability and ergodic capacity of the FD relaying satellite sensor network are derived by considering residual loop interference, channel statistical property, propagation loss, geometric satellite antenna pattern, and terminal elevation angle. Simulation results show the achieved performance gains of a full-duplex relaying satellite sensor network over traditional half-duplex relaying, and highlight the impact of key system parameters on the performance of the considered FD relaying satellite sensor network.

## 1. Introduction

### Satellite Systems

Sensor nodes are commonly involved in an extensive range of applications in environmental sensing, remote health monitoring, environmental monitoring and target tracking [1,2]. Due to the potential in providing wide coverage and high transmission rate, satellite sensor networks have been regarded as an effective approach to provide telecommunication and multimedia services for users who are separated far away, especially when the line-of-sight (LoS) link is unavailable because of the masking effect [3,4,5]. Satellite relaying is known as a basic type for communication, tracking and data exchange for future integrated satellite-terrestrial scenarios [6]. Generally, the basic architectures of satellite relaying includes two widely adopted schemes, namely, amplify-and-forward (AF) and decode-and-forward (DF) relay protocols [7]. The DF satellite relaying performs on-board processing to demodulate the received signals over the uplink, and then remodulate signals over the downlink [8]. The AF satellite relaying, which amplifies the signal transmitted from source and forwards it to the destination with fixed or channel state information (CSI) assisted gain, is of particular interest due to implementation simplicity [9,10].

Recently, due to the fact that future wireless networks require enhanced spectral efficiency to support the increasing demand of access and application services, the full-duplex (FD) techniques have received considerable attention as promising candidates for its capability in alleviating the spectrum shortage [11,12]. FD transmission mode has been widely applied in a variety of wireless systems, including vehicular networks, device-to-device (D2D) networks and multiple antenna networks [13]. However, traditional transmission scenarios in satellite sensor systems were all assumed to be in a half-duplex (HD) mode, which resulted in spectral deficiency and low on-board resource utilization. To meet the increasing demand for higher throughput in future satellite networks, several potential techniques have been considered to improve the spectral efficiency in satellite communications, among which, full-duplex (FD) mode has been considered as a promising technique since it can afford simultaneously transmission and reception on the same carrier frequency [14,15]. Until now, the work in [14] presented initial considerations on the coexistence of simultaneous transmission and reception in full-duplex satellite relaying, and justified the effectiveness of analog and digital cancellation together with the passive suppression in achieving an enhanced self-interference (SI) cancellation. The authors of [15] modeled and analyzed the signal and SI components in the context of in-band FD satellite relaying, which illustrated the feasibility and perspective of FD-enabled satellite relaying from the technical point of view. Moreover, the implementations of joint analog and digital cancellations with respect to the on-board residual SI for FD satellite relaying was explored in [16]. Although the aforemention works established fundamentals and reviewed the potential applications of FD in satellite communication systems, no contributions have been dedicated to analyzing the key performance metrics of full-duplex satellite relaying systems; thus, its performance gains and superiority over the conventional HD mode also remained unknown.

In this paper, we investigate the performance of the satellite sensor networks with full-duplex relaying under residual SI. Specifically, the main contribution presented in this work can be summarized in the following aspects:We first develop an FD-enabled framework for satellite relaying systems by applying the standard recommendations and considering practical antenna geometries, configurations, and channel characteristics. The new diagram of this paper establishes the foundation for system performance evaluation, which can be viewed as a general and extensively applicable model for various scenarios. The employed gamma distribution to approximate the log-normal distribution can result in a suitable statistical model with the same performance for practical interests, and is applicable for a variety of applications in different frequency bands including UHF-band, S-band, L-band, Ku-band and Ka-band.Our theoretical derivations provide new analytical expressions for the performance merits of outage probability and ergodic capacity of the FD satellite relaying network, which are general and applicable to cases involving arbitrary channel conditions and system parameters. To the best of our knowledge, this is the first time that such analytical expressions are developed for FD satellite relay systems, which provide an efficient and comprehensive approach to evaluate the considered system performance.The representative simulations and comparisons are provided, which clearly reveal the effects of residual SI, channel statistical property, propagation loss, geometric satellite antenna pattern, and terminal elevation angle on the system performance. Our findings indicate that a full-duplex relaying satellite sensor network can achieve a higher capacity than that of traditional half-duplex relaying.

The rest of the paper is organized as follows. In Section 2, we introduce the system and transmission model of a full-duplex relaying satellite sensor network. In Section 3, we derive the closed-form expression for the outage probability and ergodic capacity of the considered FD relaying satellite sensor network. Numerical simulations are given in Section 4. Finally, conclusions are drawn in Section 5.

**Notations**: Bold uppercase and lowercase letters represent the matrices and vectors, respectively. ${\left(\cdot \right)}^{H}$ stands for conjugate transpose operator. ${\left|\cdot \right|}^{2}$ denotes the power gain of a constant, $\left|\cdot \right|$ denotes the absolute value of a complex scalar. $E\left[\cdot \right]$ represents the expectation operator, $\exp\left(\cdot \right)$ denotes the exponential function. $CN\left(a,b\right)$ the complex Gaussian distribution of mean a and covariance *b*.

## 2. System Model

As shown in Figure 1, a sensor node is denoted as earth station (ES)-1 sends signal $x\left[i\right]$ to a remote data center or exchange center denoted as ES-2 with the help of a satellite relaying (The satellite is considered to be of transparent manner, which the bent-pipe type satellite amplifies the received signals from ES by a gain factor, and then retransmit to intended terminal [8,9,10]). Particularly, the considered scenario involves the FD mode on the uplink (ES-1 to satellite) and downlink (satellite to ES-2). The return path from user terminal (UT), i.e., ES-*i*, to gateway (GW) can also be implemented in a similar FD manner (Although some mechanisms (credited as Full Duplex) have been proposed to increase the spectrum reuse with different frequencies for the transmission and reception, the full Duplex concept in this paper always refer to in-band full duplex, that is, the ability to transmit and receive simultaneously on the same frequency band [14,15]. Under the consideration of residual interference on-board the satellite, the performance degradation requires further investigation to be quantitatively determined for the guidance of system design and performance evaluation, which will be the topic of our future research.) The Geostationary satellite (GEO) system is considered in this work, while the Doppler effect is not within the scope of this research in this work.

When the relay operates in FD mode, it concurrently receives a desired signal $x\left[i\right]$ along with loop interference $t\left[i\right]$ as (The on-board residual loop interference is taken into account due to imperfect cancellation [17]. A simple extrapolation of the radio astronomy mask shows that the admissible emission in the satellite uplink RX band would be way too high with respect to the received signal magnitude. This out-of-band leakage coming from the transmitter reduces the sensitivity of the receiver [14]. The self-interference between the on-board satellite antennas for transmitting and receiving antennas can be accordingly alleviated with directional antenna while both the complementary analog and digital active cancellation units are still required to maintain the SI under acceptable level [14,15].)
(1)$$\begin{array}{c} r\left[i\right]=\sqrt{P_{1}}h_{1}x\left[i\right]+\sqrt{P_{2}}h_{I}t\left[i\right]+n_{1}^{}\left[i\right], \end{array}$$
where $P_{1}$ and $P_{2}$ denotes the transmit power at the ES-1 and satellite, respectively. $h_{1}$ is the channel coefficient between ES-1 and satellite, $h_{I}$ is the resudial loop interference channel coefficient due to imperfect SI cancellation, and $n_{1}^{}\left[i\right]$ is the noise term satisfying $E\left\{{\left|n_{1}^{}\left[i\right]\right|}_{}^{2}\right\}=\sigma_{1}^{2}$.

Then, the relay amplifies the received signal by a gain factor β, which introduces a delay of τ as
(2)$$\begin{array}{c} \sqrt{P_{2}}t\left[i\right]=\beta r\left[i-\tau \right]. \end{array}$$

By combining (1) and (2), and considering the FD concept, the sum of the received signal and loop interference at the satellite relaying with respect to infinite time slots with an iterative manner can be obtained as (see system model in [17] and [18])
(3)$$\begin{array}{cc} \sqrt{P_{2}}t\left[i\right] & =\beta \sum_{j=1}^{\infty }{\left(h_{I}^{}\beta \right)}_{}^{j-1}\\ \; & \times \left(\sqrt{P_{1}}h_{1}^{}x\left[i-j\tau \right]\;+\;n_{1}^{}\left[i-j\tau \right]\right). \end{array}$$

Thus, the output signal power is calculated from (3) as
(4)$$\begin{array}{cc} P_{2}E\left\{{\left|t\left[i\right]\right|}_{}^{2}\right\} & =\beta_{}^{2}\sum_{j=1}^{\infty }{\left({\left|h_{I}^{}\right|}_{}^{2}\beta_{}^{2}\right)}_{}^{j-1}\left(P_{1}{\left|h_{1}^{}\right|}_{}^{2}+\sigma_{1}^{2}\right)\\ \; & =\beta_{}^{2}\frac{P_{1}{\left|h_{1}^{}\right|}_{}^{2}+\sigma_{1}^{2}}{1-{\left|h_{I}^{}\right|}_{}^{2}\beta_{}^{2}}. \end{array}$$

By considering the normalized condition $E\left\{{\left|t\left[i\right]\right|}_{}^{2}\right\}=1$ to guarantee the transmit power of satellite relaying within the prescribed bound (In particular, we consider the normalized conditions $E\left\{{\left|x\left[i\right]\right|}_{}^{2}\right\}=1$ and $E\left\{{\left|t\left[i\right]\right|}_{}^{2}\right\}=1$ over the DVB-S2 constellations [12].), the gain factor is obtained by satisfying (The scaling factor β is to ensure that the power transmitted from the satellite remains within the prescribed bounds [15].)
(5)β=P2P2P1h12+P2hI2+σ12P1h12+P2hI2+σ121/2.

Finally, the received signal at the ES-2 can be expressed as
(6)$$\begin{array}{c} y\left[i\right]=\sqrt{P_{2}}h_{2}t\left[i\right]+n_{2}^{}\left[i\right], \end{array}$$
where $h_{2}$ denotes the channel coefficient between satellite and ES-2, and $n_{2}^{}\left[i\right]$ is the noise component obeying $E\left\{{\left|n_{2}^{}\left[i\right]\right|}_{}^{2}\right\}=\sigma_{2}^{2}$. From (6), the instantaneous received power can be derived as
(7)$$\begin{array}{c} \begin{array}{c} E\left\{{\left|y\left[i\right]\right|}_{}^{2}\right\}=\beta_{}^{2}P_{1}{\left|h_{1}^{}\right|}_{}^{2}{\left|h_{2}^{}\right|}_{}^{2}\\ \;+\;\beta_{}^{2}\left(P_{1}{\left|h_{1}^{}\right|}_{}^{2}\;+\;\sigma_{1}^{2}\right){\left|h_{2}^{}\right|}_{}^{2}\frac{\beta_{}^{2}{\left|h_{I}^{}\right|}_{}^{2}}{1-\beta_{}^{2}{\left|h_{I}^{}\right|}_{}^{2}}\;+\;\beta_{}^{2}{\left|h_{2}^{}\right|}_{}^{2}\sigma_{1}^{2}\;+\;\sigma_{2}^{2}. \end{array} \end{array}$$

To this end, by reformulating the signal, residual loop interference and noise power in (7), the received signal-to-interference and noise ratio (SINR) can be expressed as
(8)γ=P1h12h22P1h12+σ12hI211β2β2−hI2+σ12h22+σ22β2.

Although some slow fluctuation would happen during on-board processing, but most of the channel components would be still static [15,18,19], by substituting (5) into (8) and involving the common assumption of non-fading residual interference channel, (8) can be further simplified as
(9)$$\begin{array}{c} \gamma_{}=\frac{\gamma_{1}^{}\gamma_{2}^{}}{\gamma_{1}^{}+\left(\gamma_{2}^{}+1\right)\left({\bar{\gamma }}_{I}^{}+1\right)}, \end{array}$$
where signal-to-noise (SNR) level of uplink and downlink are γ1=P1h12P1h12σ12σ12, γ2=P2h22P2h22σ22σ22, respectively. The residual interference level ${\bar{\gamma }}_{I}^{}=E\left[\gamma_{I}^{}\right]$ follows the well-known principles of echo cancellation, which can be expected to come from the complementary analog and digital cancellation efforts.

To determine the on-board magnitude of the achievable SI cancellation levels, it is considered that the SI cancellation circuit is capable of matching the delay perfectly [14]. Hence, the remaining error component is introduced by the non-ideal estimated weight of the coupled SI signal. Due to the fact that $h_{I}$ determines the strength of the SI, it can be explained as the residual SI channel after certain SI cancellation measures are applied. Considering the related true and time-invariant nature, the SI channel residual echo can be expressed as [14,15]
(10)$$\begin{array}{c} e=h_{I}-{\tilde{h}}_{I}=h_{I}-\left(1-\varepsilon \right)h_{I}, \end{array}$$
where ε denotes the relative estimation error, and the echo cancellation performance can be measured as [15]
(11)$$\begin{array}{c} \mathrm{Cancellation}[\mathrm{dB}]=10\log_{10}\left(\frac{E\left[{\left|h_{I}^{}\right|}_{}^{2}\right]}{E\left[{\left|e\right|}_{}^{2}\right]}\right), \end{array}$$

Based on the relationship in (10), (11) is equal to
(12)$$\begin{array}{c} \mathrm{Cancellation}[\mathrm{dB}]=-20\log_{10}\left(\left|\varepsilon \right|\right) \end{array}$$

As can be observed in (12), the cancellation performance degrades significantly even with low errors in the estimation of the coupling magnitude, which can be shown in Table 1 (As proved in [19] for the reference conceived to exchange information between two ends through a satellite, the on-borad digital cancellation can target cancellation levels about 30 dB in a concept known commercially as Paired-Carrier Multiple Access or DoubleTalk Carrier-in-Carrier.).

## 3. Performance Evaluation

In this section, we derive the closed-from expressions for the key performance merits, i.e., outage probability [20] and ergodic capacity [21] of the considered full-duplex satellite relaying systems. Before delving into the detailed analysis, the satellite channel is firstly presented in what follows.

### 3.1. Satellite Channel Model

By considering the on-board beam gain, pathloss, channel fading and ES antenna gain, $h_{i},\left(i\in \left\{1,2\right\}\right)$ can be wirtten as [22]
(13)$$\begin{array}{c} h_{i}^{}=G_{i}^{\frac{1}{2}}g_{i}^{}, \end{array}$$
with [23]
(14)$$\begin{array}{c} G_{i}^{}=LG_{U}^{}G_{S,\max}^{}{\left(\frac{J_{1}\left(u_{i}^{}\right)}{2u_{i}^{}}+36\frac{J_{3}\left(u_{i}^{}\right)}{u_{i}^{3}}\right)}_{}^{2}, \end{array}$$
where the above parameters are defined below [24]:$L_{}^{}$: Free space loss between for the uplink and downlink computed as
$$\begin{array}{c} L={\left(\frac{c}{4\pi df_{c}}\right)}_{}^{2}, \end{array}$$
where *c* is the propagation speed, $f_{c}$ the frequency and *d* the distance, and *d* = 35,786 km.$G_{U}^{}$: Antenna gain at the ES-$i,\left(i\in \left\{1,2\right\}\right)$.$G_{S,\max}^{}$: Maximum beam gain at the on-board antenna boresight.$u_{i}$: Auxiliary variable in determining the on-board beam gain factor for a given a ES’s position, which is defined as [25]
(15)$$u_{i}=2.07123\frac{\sin\varphi_{i}}{\sin\varphi_{3\mathrm{dB}}^{}},$$
where $\varphi_{i}$ and $\varphi_{3\mathrm{dB}}$ represents the beam angle with respect to the beam center and 3-dB angle, respectively.$g_{i}^{}$: Fading channel coefficient of the satellite links. Among the different atmosphere effects, rain attenuation is regarded as the major impairment which is commonly described as a log-normal distribution. However, in a practical scenario of the existing literature, the application of log-normal distribution in modeling the shadowing fading would lead to a quite complicated expressions for characterizing the key merits of both the first- and second-order statistical properties [26]. On the other hand, the shadowed-Rician model proposed originally in [26], which adopts the gamma distribution to approximate the log-normal distribution, can result in a simpler form for channel statistics with the similar performance for practical cases. As can be found in the existing literature, the shadowed-Rician distribution can be applied in difference frequency bands, including UHF-band, L-band, Ku-band, Ka-band and etc [7,8,9]. Under this situation, this paper has employed an alternative approach for atmosphere and weather effects according to the existing references, which can be applied to both the fixed and mobile terminals operating in various propagation environments [7,8,9,23,24,25]. Accordingly, the probability density function (PDF) of the channel gain ${\left|g_{i}^{}\right|}_{}^{2}$ can be given by [26]
(16)$$\begin{array}{c} f_{{\left|g_{i}^{}\right|}_{}^{2}}^{}\left(x\right)=\alpha_{i}\exp\left(-\beta_{i}x\right){}_{1}F_{1}\left(m_{i};1;\delta_{i}x\right), \end{array}$$
where ${}_{1}F_{1}\left(a;b;c\right)$ denotes the confluent hypergeometric function ([27], Equation (9.210.1)). Based on the results in [26], the related parameters $\alpha_{i}$, $\beta_{i}$ and $\delta_{i}$ can be, respectively, calculated by the following identities
(17a)αi=2bimi2bimi2bimi+Ωi2bimi+Ωimi2bimi2bimi2bimi+Ωi2bimi+Ωimi2bi2bi,
(17b)βi=112bi2bi,
(17c)δi=ΩiΩi2bi2bimi+Ωi2bi2bimi+Ωi,
where $\Omega_{i}$ denotes the average power in terms of the LoS component, $2b_{i}$ stands for the multipath average power, and $m_{i}$ represents the Nakagami-*m* fading severity. As illustrated in [26], the channel parameters $b_{i}$, $m_{i}$ and $\Omega_{i}$ of the satellite links can be determined with respect to the elevation angles $\theta_{i}^{}$, which can be calculated by the following identities over the range $20^{\circ }\leq \theta_{i}\leq 80^{\circ }$ as
(18)$$\begin{array}{cc} b_{i}\left(\theta_{i}\right)\;= & \;\;-\;4.7943\times 10^{-8}\theta_{i}^{3}\;+\;5.5784\times 10^{-6}\theta_{i}^{2}\\ \; & -2.1344\times 10^{-4}\theta_{i}+3.2710\times 10^{-2}\\ m_{i}\left(\theta_{i}\right)\;= & \;6.3739\times 10^{-5}\theta_{i}^{3}\;+\;5.8533\times 10^{-4}\theta_{i}^{2}\\ \; & -1.5973\times 10^{-1}\theta_{i}+3.5156\\ \Omega_{i}\left(\theta_{i}\right)\;= & \;1.4428\times 10^{-5}\theta_{i}^{3}\;-\;2.3798\times 10^{-3}\theta_{i}^{2}\\ \; & +1.2702\times 10^{-1}\theta_{i}-1.4864 \end{array}.$$
where the frequent heavy shadowing (FHS), average shadowing (AS) and infrequent light shadowing (IFL) scenarios can be precisely simulated corresponding to the low, medium and high elevation angles [26].

### 3.2. Outage Probability

The quality of satellite services can be determined by using the outage probability, which is commonly defined as the probability that the received SINR falls below a predefined threshold $\gamma_{th}$, namely [28]
(19)$$\begin{array}{c} P_{out}=\mathrm{Pr}\left(\gamma \leq \gamma_{th}\right)=F_{\gamma_{}}\left(\gamma_{th}\right), \end{array}$$
where $F_{\gamma_{}}\left(\cdot \right)$ denotes the cumulative distribution function (CDF) of γ. From (9), $F_{\gamma_{}}\left(x\right)$ can be expressed as
(20)$$\begin{array}{c} F_{\gamma_{}}\left(x\right)=\int_{0}^{\left({\bar{\gamma }}_{I}^{}+1\right)\gamma_{th}}f_{\gamma_{1}}\left(y\right)dy\int_{0}^{\infty }f_{\gamma_{2}}\left(x\right)dx\;\;\;\;\;\;\;\;\\ +\int_{\left({\bar{\gamma }}_{I}^{}+1\right)\gamma_{th}}^{\infty }F_{\gamma_{2}}\left(\frac{\left(y+{\bar{\gamma }}_{I}^{}+1\right)\gamma_{th}}{y-\left({\bar{\gamma }}_{I}^{}+1\right)\gamma_{th}}\right)f_{\gamma_{1}}\left(y\right)dy. \end{array}$$

For simplicity, by assuming the Nakagami parameter $m_{i}$ in (13) takes on integer values [21,23], and defining ${\bar{\gamma }}_{i}^{}=\frac{P_{1}G_{i}^{}}{\sigma_{i}^{2}}$, $f_{\gamma_{i}^{}}^{}\left(x\right)$ can be simplified as
(21)$$\begin{array}{c} f_{\gamma_{i}^{}}^{}\left(x\right)=\alpha_{i}\exp\left(-\frac{\left(\beta_{i}^{}-\delta_{i}^{}\right)}{{\bar{\gamma }}_{i}^{}}x\right)\sum_{k_{i}=0}^{m_{i}^{}-1}\Lambda \left(k_{i},{\bar{\gamma }}_{i}^{}\right)x_{}^{k_{i}^{}}, \end{array}$$
with
$$\begin{array}{c} \Lambda \left(k_{i},{\bar{\gamma }}_{i}^{}\right)=\frac{{\left(-1\right)}_{}^{k_{i}}{\left(1-m_{i}^{}\right)}_{k}^{}\delta_{i}^{k_{i}^{}}}{{\left(k_{i}^{}!\right)}_{}^{2}{\bar{\gamma }}_{i}^{k_{i}^{}+1}}. \end{array}$$

Then, by applying ([27], Equation (3.351.2)) along with integration computation, $F_{\gamma_{i}^{}}^{}\left(x\right)$ can be obtained as
(22)$$\begin{array}{cc} F_{\gamma_{i}^{}}^{}\left(x\right) & =\int_{0}^{x}f_{\gamma_{i}^{}}^{}\left(y\right)dy\\ \; & =1-\alpha_{i}\exp\left(-\frac{\left(\beta_{i}^{}\;-\;\delta_{i}^{}\right)x}{{\bar{\gamma }}_{i}^{}}\right)\sum_{k_{i}^{}=0}^{m_{i}^{}-1}\sum_{l_{i}^{}=0}^{k_{i}^{}}\Xi \left(k_{i},l_{i},{\bar{\gamma }}_{i}^{}\right)x_{}^{l_{i}^{}}, \end{array}$$
(23)$$\begin{array}{cc} F_{\gamma_{}}\left(x\right) & =1-\alpha_{1}\alpha_{2}\sum_{k_{1}^{}=0}^{m_{1}^{}-1}\Lambda \left(k_{1},{\bar{\gamma }}_{1}^{}\right)\sum_{k_{2}^{}=0}^{m_{2}^{}-1}\sum_{l_{2}^{}=0}^{k_{2}}\Xi \left(k_{2}^{},l_{2}^{},{\bar{\gamma }}_{2}^{}\right)\\ \; & \times \int_{\left({\bar{\gamma }}_{I}^{}+1\right)\gamma_{th}}^{\infty }y_{}^{k_{1}^{}}{\left(\frac{\left(y+{\bar{\gamma }}_{I}^{}+1\right)\gamma_{th}}{y-\left({\bar{\gamma }}_{I}^{}+1\right)\gamma_{th}}\right)}_{}^{l_{2}^{}}\exp\left(-\frac{\left(\beta_{2}^{}-\delta_{2}^{}\right)}{{\bar{\gamma }}_{2}^{}}\frac{\left(y+{\bar{\gamma }}_{I}^{}+1\right)\gamma_{th}}{y-\left({\bar{\gamma }}_{I}^{}+1\right)\gamma_{th}}-\frac{\left(\beta_{1}^{}-\delta_{1}^{}\right)}{{\bar{\gamma }}_{1}^{}}y\right)dy, \end{array}$$
(24)$$\begin{array}{c} F_{\gamma_{}}\left(x\right)=1-2\alpha_{1}\alpha_{2}\sum_{k_{1}^{}=0}^{m_{1}^{}-1}\Lambda \left(k_{1},{\bar{\gamma }}_{1}^{}\right)\sum_{k_{2}^{}=0}^{m_{2}^{}-1}\sum_{l_{2}^{}=0}^{k_{2}}\Xi \left(k_{2}^{},l_{2}^{},{\bar{\gamma }}_{2}^{}\right)\exp\left(-\left(B+\mathrm{AC}\right)\gamma_{th}\right)\;\;\;\;\;\;\;\;\;\\ \times \sum_{i=0}^{k_{1}^{}}\left(\begin{array}{c} k_{1}^{}\\ i \end{array}\right)\sum_{j=0}^{l_{2}^{}}\left(\begin{array}{c} l_{2}^{}\\ j \end{array}\right)\frac{C_{}^{k_{1}^{}-i+j}D_{}^{j}}{\gamma_{th}^{i-k_{1}^{}-l_{2}^{}}}{\left(\frac{\mathrm{BCD}\gamma_{th}^{}}{A}\right)}^{\frac{i-j+1}{2}}K_{i-j+1}\left(2\sqrt{\mathrm{ABCD}\gamma_{th}^{}}\right), \end{array}$$
with
$$\begin{array}{c} \Xi \left(k_{i}^{},l_{i}^{},{\bar{\gamma }}_{i}^{}\right)=\frac{{\left(-1\right)}_{}^{k_{i}^{}}{\left(1-m_{i}^{}\right)}_{k_{i}^{}}^{}\delta_{i}^{k_{i}^{}}}{k_{i}^{}!l_{i}^{}!{\left(\beta_{i}^{}-\delta_{i}^{}\right)}_{}^{k_{i}^{}-l_{i}^{}+1}{\bar{\gamma }}_{i}^{l_{i}^{}}}. \end{array}$$

Substituting (21) and (22) into (20), we get (23) as shown on the top of next page. Then, by letting $z=y-\left({\bar{\gamma }}_{I}^{}+1\right)\gamma_{th}$, and $A=\frac{\left(\beta_{1}^{}-\delta_{1}^{}\right)}{{\bar{\gamma }}_{1}^{}}$, $B=\frac{\left(\beta_{2}^{}-\delta_{2}^{}\right)}{{\bar{\gamma }}_{2}^{}}$, $C={\bar{\gamma }}_{I}^{}+1$, $D=\gamma_{th}^{}+1$ and applying ([27], Equation (3.471.9)), the closed-form expression of $F_{\gamma_{}}\left(x\right)$ can be derived as (21). To this end, by replacing *x* with $\gamma_{th}$ in (24), it is straightforward to calculate the OP of the FD satellite relaying system.

### 3.3. Ergodic Capacity

The ergodic capacity is defined as the expectation of the instantaneous mutual information between the end-to-end SINR, which can be expressed as [29,30]
(25)$$\begin{array}{c} C_{}^{}=E\left[\log_{2}\left(1+\gamma \right)\right], \end{array}$$
where the pre-log factor equals to 1 due to no spectral loss in FD mode. By utilizing (9) into (25) and denoting ${\tilde{\gamma }}_{1}^{}=\frac{\gamma_{1}^{}}{{\bar{\gamma }}_{I}^{}+1}$, $C_{}^{}$ can be written as
(26)$$\begin{array}{cc} C_{}^{} & =E\left[\log_{2}\left(1+\frac{\gamma_{1}^{}\gamma_{2}^{}}{\gamma_{1}^{}+\left(\gamma_{2}^{}+1\right)\left({\bar{\gamma }}_{I}^{}+1\right)}\right)\right]\\ \; & =E\left[\log_{2}\left(\frac{\left(1+{\tilde{\gamma }}_{1}\right)\left(1+\gamma_{2}\right)}{{\tilde{\gamma }}_{1}+\gamma_{2}+1}\right)\right]\\ \; & =C_{1}^{}+C_{2}^{}-C_{3}^{}, \end{array}$$
where $C_{1}$, $C_{2}$ and $C_{3}$ can be, respectively, expressed as
(27a)$$\begin{array}{c} C_{1}^{}=E\left[\log_{2}\left(1+{\tilde{\gamma }}_{1}\right)\right],\;\;\; \end{array}$$
(27b)$$\begin{array}{c} C_{2}^{}=E\left[\log_{2}\left(1+\gamma_{2}\right)\right],\;\;\; \end{array}$$
(27c)$$\begin{array}{c} C_{3}^{}=E\left[\log_{2}\left(1+{\tilde{\gamma }}_{1}+\gamma_{2}\right)\right]. \end{array}$$

To begin with, we first turn to drive the analytical results of $C_{1}$ and $C_{1}$ as
(28)$$\begin{array}{c} C_{1}^{}=\frac{1}{\ln2}\int_{0}^{\infty }\ln\left(1+x\right)f_{{\tilde{\gamma }}_{1}}^{}\left(x\right)dx, \end{array}$$
and
(29)$$\begin{array}{c} C_{2}^{}=\frac{1}{\ln2}\int_{0}^{\infty }\ln\left(1+x\right)f_{\gamma_{2}}^{}\left(x\right)dx. \end{array}$$

Then, using (21) into (28) and (29), and applying ([31], Equation (Equation 11)) to express $\ln\left(1+x\right)$ in terms of Meijer-G function as
(30)$$\begin{array}{c} \ln\left(1+x\right){=G}_{2,2}^{1,2}\left[x\left|\begin{array}{c} 1,1\\ 1,0 \end{array}\right.\right], \end{array}$$
we get the analytical results of $C_{1}$ and $C_{2}$ as
(31)$$\begin{array}{c} C_{1}^{}=\frac{\alpha_{1}}{\ln2}\sum_{k_{1}=0}^{m_{1}^{}-1}\frac{\Lambda \left(k_{1},{\tilde{\gamma }}_{1}^{}\right)}{{\left(AC\right)}_{}^{k_{1}^{}+1}}G_{2,2}^{1,2}\left[AC\left|\begin{array}{c} -k_{1}^{},1,1\\ 1,0 \end{array}\right.\right], \end{array}$$
and
(32)$$\begin{array}{c} C_{2}^{}=\frac{\alpha_{2}}{\ln2}\sum_{k_{2}=0}^{m_{2}^{}-1}\frac{\Lambda \left(k_{2},\gamma_{2}^{}\right)}{B_{}^{k_{2}^{}+1}}G_{2,2}^{1,2}\left[B\left|\begin{array}{c} -k_{2}^{},1,1\\ 1,0 \end{array}\right.\right]. \end{array}$$

In deriving (30) and (31), we have applied ([27], Equation (7.813.1)). Due to the fact that the closed-form PDF expression of ${\tilde{\gamma }}_{1}+\gamma_{2}$ is mathematically intractable, we employ an alternative approach based on the moment generating function (MGF) to derive $C_{3}$ as
(33)$$\begin{array}{c} C_{3}^{}=\frac{1}{\ln2}\int_{0}^{\infty }Ei\left(-s\right)M_{\gamma_{3}^{}}^{\left(1\right)}\left(s\right)ds, \end{array}$$
where $M_{\gamma_{3}^{}}^{\left(1\right)}\left(s\right)$ denotes the first-order derivation with respect to *s*. Due to the independent nature of ${\tilde{\gamma }}_{1}$ and $\gamma_{2}$, we have the following property of $M_{\gamma_{3}^{}}^{}\left(s\right)$ as
(34)$$\begin{array}{c} M_{\gamma_{3}^{}}^{}\left(s\right)=M_{{\tilde{\gamma }}_{1}^{}}^{}\left(s\right)M_{\gamma_{2}^{}}^{}\left(s\right), \end{array}$$
and then the $M_{\gamma_{3}^{}}^{\left(1\right)}\left(s\right)$ can be expressed as
(35)$$\begin{array}{cc} M_{\gamma_{3}^{}}^{\left(1\right)}\left(s\right) & ={\left[M_{{\tilde{\gamma }}_{1}^{}}^{}\left(s\right)M_{\gamma_{2}^{}}^{}\left(s\right)\right]}_{}^{\left(1\right)}\\ \; & =M_{{\tilde{\gamma }}_{1}^{}}^{\left(1\right)}\left(s\right)M_{\gamma_{2}^{}}^{}\left(s\right)+M_{{\tilde{\gamma }}_{1}^{}}^{}\left(s\right)M_{\gamma_{2}^{}}^{\left(1\right)}\left(s\right). \end{array}$$

According to the definition of MGF, $M_{{\tilde{\gamma }}_{1}^{}}^{}\left(s\right)$ and $M_{\gamma_{2}^{}}^{}\left(s\right)$ can be derived as
(36)$$\begin{array}{c} M_{{\tilde{\gamma }}_{1}^{}}^{}\left(s\right)=\alpha_{1}\sum_{k_{1}=0}^{m_{1}^{}-1}\Lambda \left(k_{1},{\tilde{\gamma }}_{1}^{}\right)\frac{k_{1}^{}!}{{\left(s+AC\right)}_{}^{k_{1}^{}+1}}, \end{array}$$
and
(37)$$\begin{array}{c} M_{\gamma_{2}^{}}^{}\left(s\right)=\alpha_{2}\sum_{k_{2}=0}^{m_{2}^{}-1}\Lambda \left(k_{2},{\bar{\gamma }}_{2}^{}\right)\frac{k_{2}^{}!}{{\left(s+B\right)}_{}^{k_{2}^{}+1}}. \end{array}$$

Then, by differentiating (33) and (34) with respect to *s*, the analytical results of $M_{{\tilde{\gamma }}_{1}^{}}^{\left(1\right)}\left(s\right)$ and $M_{\gamma_{2}^{}}^{\left(1\right)}\left(s\right)$ can be obtained as
(38)$$\begin{array}{c} M_{{\tilde{\gamma }}_{1}^{}}^{\left(1\right)}\left(s\right)\;=\;-\alpha_{1}\sum_{k_{1}=0}^{m_{1}^{}-1}\Lambda \left(k_{1},{\tilde{\gamma }}_{1}^{}\right)\frac{\left(k_{1}^{}+1\right)!}{{\left(s+AC\right)}_{}^{k_{1}^{}+2}}, \end{array}$$
and
(39)$$\begin{array}{c} M_{\gamma_{2}^{}}^{\left(1\right)}\left(s\right)=-\alpha_{2}\sum_{k_{2}=0}^{m_{2}^{}-1}\Lambda \left(k_{2},{\bar{\gamma }}_{2}^{}\right)\frac{\left(k_{2}^{}+1\right)!}{{\left(s+B\right)}_{}^{k_{2}^{}+2}}. \end{array}$$

Subsequently, by combining (33) and (35), $C_{3}^{}$ can be further expressed as
(40)$$\begin{array}{c} C_{3}^{}=\frac{1}{\ln2}\left[\int_{0}^{\infty }Ei\left(-s\right)M_{{\tilde{\gamma }}_{1}^{}}^{\left(1\right)}\left(s\right)M_{\gamma_{2}^{}}^{}\left(s\right)ds\right.\\ \left.+\int_{0}^{\infty }Ei\left(-s\right)M_{{\tilde{\gamma }}_{1}^{}}^{}\left(s\right)M_{\gamma_{2}^{}}^{\left(1\right)}\left(s\right)ds\right]. \end{array}$$
(41)$$\begin{array}{cc} C_{3}^{} & =\frac{\alpha_{1}\alpha_{2}}{\ln2}\sum_{k_{1}=0}^{m_{1}^{}-1}\frac{\Lambda \left(k_{1},{\tilde{\gamma }}_{1}^{}\right)}{{\left(\left({\bar{\gamma }}_{I}^{}+1\right)A\right)}_{}^{k_{1}^{}+1}}\sum_{k_{2}=0}^{m_{2}^{}-1}\frac{\Lambda \left(k_{2},{\bar{\gamma }}_{2}^{}\right)}{B_{}^{k_{2}^{}+1}}\left[\frac{1}{\left({\bar{\gamma }}_{I}^{}+1\right)A}\right.\\ \; & \times \underset{I_{1}^{}}{\underset{︸}{\int_{0}^{\infty }G_{1,1}^{1,1}\left[\left({\bar{\gamma }}_{I}^{}+1\right)As\left|\begin{array}{c} -k_{1}^{}-1\\ 0 \end{array}\right.\right]G_{1,1}^{1,1}\left[Ax\left|\begin{array}{c} -k_{2}^{}\\ 0 \end{array}\right.\right]G_{1,2}^{2,0}\left[s\left|\begin{array}{c} 1\\ 0,0 \end{array}\right.\right]ds}}\\ \; & +\left.\underset{I_{2}^{}}{\underset{︸}{\frac{1}{A_{}^{k_{2}^{}+1}}\int_{0}^{\infty }G_{1,1}^{1,1}\left[\left({\bar{\gamma }}_{I}^{}+1\right)As\left|\begin{array}{c} -k_{1}^{}\\ 0 \end{array}\right.\right]G_{1,1}^{1,1}\left[Ax\left|\begin{array}{c} -k_{2}^{}-1\\ 0 \end{array}\right.\right]G_{1,2}^{2,0}\left[s\left|\begin{array}{c} 1\\ 0,0 \end{array}\right.\right]ds}}\right], \end{array}$$

Then, substituting (36)–(39) into (40) yields (41). To solve the associated multiple integrals, we then apply the following Meijer-G function representations
(42)$$\begin{array}{c} {\left(1+\alpha x\right)}_{}^{\beta }=\frac{1}{\Gamma \left(-\beta \right)}G_{1,1}^{1,1}\left[\alpha x\left|\begin{array}{c} \beta +1\\ 0 \end{array}\right.\right], \end{array}$$
and
(43)$$\begin{array}{c} Ei\left(-s\right)=-G_{1,2}^{2,0}\left[s\left|\begin{array}{c} 1\\ 0,0 \end{array}\right.\right], \end{array}$$
along with ([32], Equation (3.1)), and derive the closed-form expression of $C_{3}$ as (44).

Finally, by substituting (31), (32) and (40) into (26), one can directly calculate the ergodic capacity of the FD satellite relaying system as (45).

(44)$$\begin{array}{cc} C_{3}^{} & =\frac{\alpha_{1}\alpha_{2}}{\ln2}\sum_{k_{1}=0}^{m_{1}^{}-1}\frac{\Lambda \left(k_{1},{\tilde{\gamma }}_{1}^{}\right)}{{\left(AC\right)}_{}^{k_{1}^{}+2}}\sum_{k_{2}=0}^{m_{2}^{}-1}\frac{\Lambda \left(k_{2},{\bar{\gamma }}_{2}^{}\right)}{B_{}^{k_{2}^{}+1}}\\ \; & \times \left[G_{2,[1:1],1,[1:1]}^{2,1,1,1,1}\left[\begin{array}{c} AC\\ B \end{array}\left|\begin{array}{c} 0,0\\ k_{1}^{}+2;k_{2}^{}+1\\ 2\\ 0;0 \end{array}\right.\right]+G_{2,[1:1],1,[1:1]}^{2,1,1,1,1}\left[\begin{array}{c} AC\\ B \end{array}\left|\begin{array}{c} 0,0\\ k_{1}^{}+1;k_{2}^{}+2\\ 2\\ 0;0 \end{array}\right.\right]\right]. \end{array}$$

(45)$$\begin{array}{c} C^{}=\frac{\alpha_{1}}{\ln2}\sum_{k_{1}=0}^{m_{1}^{}-1}\frac{\Lambda \left(k_{1},{\tilde{\gamma }}_{1}^{}\right)}{{\left(AC\right)}_{}^{k_{1}^{}+1}}G_{2,2}^{1,2}\left[AC\left|\begin{array}{c} -k_{1}^{},1,1\\ 1,0 \end{array}\right.\right]+\frac{\alpha_{2}}{\ln2}\sum_{k_{2}=0}^{m_{2}^{}-1}\frac{\Lambda \left(k_{2},\gamma_{2}^{}\right)}{B_{}^{k_{2}^{}+1}}G_{2,2}^{1,2}\left[B\left|\begin{array}{c} -k_{2}^{},1,1\\ 1,0 \end{array}\right.\right]\\ -\frac{\alpha_{1}\alpha_{2}}{\ln2}\sum_{k_{1}=0}^{m_{1}^{}-1}\frac{\Lambda \left(k_{1},{\tilde{\gamma }}_{1}^{}\right)}{{\left(AC\right)}_{}^{k_{1}^{}+2}}\sum_{k_{2}=0}^{m_{2}^{}-1}\frac{\Lambda \left(k_{2},{\bar{\gamma }}_{2}^{}\right)}{B_{}^{k_{2}^{}+1}}\;\;\;\;\;\;\;\;\;\;\;\;\;\;\;\;\;\;\;\;\;\;\;\;\;\;\;\;\;\;\;\;\;\;\\ \times \left[G_{2,[1:1],1,[1:1]}^{2,1,1,1,1}\left[\begin{array}{c} AC\\ B \end{array}\left|\begin{array}{c} 0,0\\ k_{1}^{}+2;k_{2}^{}+1\\ 2\\ 0;0 \end{array}\right.\right]+G_{2,[1:1],1,[1:1]}^{2,1,1,1,1}\left[\begin{array}{c} AC\\ B \end{array}\left|\begin{array}{c} 0,0\\ k_{1}^{}+1;k_{2}^{}+2\\ 2\\ 0;0 \end{array}\right.\right]\right], \end{array}$$

## 4. Numerical Results

In this section, numerical results for outage probability and ergodic capacity of the FD satellite relaying system are validated through comparison with Monte Carlo simulations. Specifically, the system parameters are provided in Table 2. Furthermore, the transmitted powers are denoted as $P_{1}=P_{2}=P$ without loss of generality [6,7,8,9].

Figure 2 shows the outage probability comparison between FD and HD satellite relaying systems for different values of ${\bar{\gamma }}_{I}$. The Monte Carlo simulations are consistent with the theoretical derivation in the presence of different residual loop interference levels, which justifies the correctness of derived analytical expression. Besides, the outage probability of the FD-based satellite relaying system is higher than that of HD-based modes. This is due to the fact that the residual loop interference at the satellite relaying poses a detrimental effect on the system performance. Bedsides, with the increase of ${\bar{\gamma }}_{I}$, the outage performance becomes worse, which justifies the necessity of SI cancellation in FD mode. From the theoretical derivations, an indication of the allowable SI levels for which the considered FD operation in satellite relaying provides performance gains over the traditional HD relaying is warranted.

Figure 3 depicts the outage probability of FD satellite relaying system for different off-boresight beam angles $\varphi_{1}=\varphi_{2}=\varphi$ and elevation angles $\theta_{1}=\theta_{1}=\theta$. It is shown that the outage performance can be improved with low values of off-boresight angle and high values of elevation angles. This is because when the ES nodes are located closer to the satellite beam center, an enhanced beam gain factor would be obtained. Moreover, the satellite links with higher elevation angles experience weaker shadowing severities, thus leading to an improved propagation quality.

Figure 4 shows the performance comparison between FD and HD modes versus residual error level *p*. The Monte Carlo simulations are consistent with the theoretical derivation in the presence of different residual loop interference levels, which justifies the correctness of derived analytical expression. Further, the analysis can also serve as benchmark on interference cancellation performance with respect to their residual error levels. As can be seen, although the SI is inevitable, the ergodic capacity of FD mode is apparently superior to the HD modes, which is obtained due to time multiplexing gain by achieving simultaneously data transmission and reception. Besides, the ergodic capacity of the FD-based satellite relaying system is superior than that of the HD-based mode. This is due to the fact that the residual loop interference at the satellite relaying poses a detrimental effect on the system performance. Bedsides, with the increase of *p*, the system performance becomes degraded, which justifies the importance of integrity and accuracy in SI cancellation for FD mode. Overall, from the theoretical derivations, the allowable SI error levels for which the considered FD operation in satellite relaying provides performance gains over the traditional HD relaying is warranted.

Figure 5 depicts the ergodic capacity comparison between FD and HD satellite relaying systems for different off-boresight beam angles. It is shown that the system performance for both FD and HD modes can be improved with a smaller off-boresight angles $\varphi_{1}$ and $\varphi_{2}$ for both uplink and downlink. This is because when the ES nodes are located closer to the satellite beam center, an enhanced beam gain factor would be obtained. However, it is worth noticing that the off-boresight angle $\varphi_{2}$ for downlink exhibits a more notable impact of the system capacity than that of $\varphi_{1}$ for uplink in FD mode. This phenomenon is because the residual loop interference would be corresponding enhanced with a higher channel gain of uplink transmission. Figure 6 illustrates the ergodic capacity comparison between FD and HD satellite relaying systems for elevation angles $\theta_{1}$ and $\theta_{2}$. As can be observed, the increasing values of $\theta_{1}$ and $\theta_{2}$ result in the improved system capacity for both modes, which demonstrates the satellite links with higher elevation angles experience weaker shadowing severities, thus leading to an improved propagation quality. Moreover, we find that the impact of $\theta_{1}$ is marginal than that of $\theta_{2}$, which can be explained by the similar fact of Figure 3.

## 5. Discussion

In this paper, we have provided a detailed performance evaluation of a satellite relaying system operating in full-duplex mode. Assuming that the loop interference can not be completely suppressed, novel closed-form expressions for the outage probability and ergodic capacity are derived, which clearly reveal the effects of time multiplexing, on-broad beam angle, residual loop interference level and ES elevation angle on the considered system. We can conclude that full-duplex satellite relaying brings a significant capacity gain besides the residual loop interference, which is obtained at the cost of a certain loss in the outage probability. The findings of this paper quantitatively analyzed the impact of key system parameters on the FD satellite relaying, and also provided an intuitive guidance for the system design, performance evaluation, and implementation principles of FD technique in satellite systems.

## Figures and Tables

**Figure 1 sensors-19-05453-f001:**
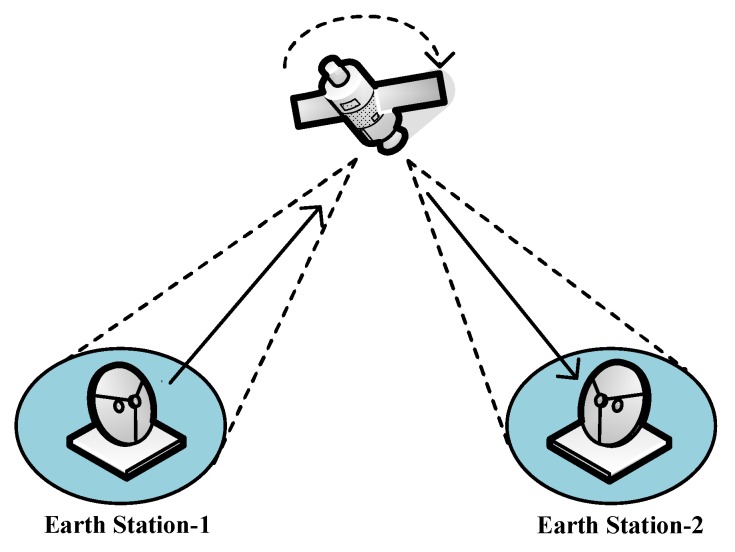
Illustration of the system model.

**Figure 2 sensors-19-05453-f002:**
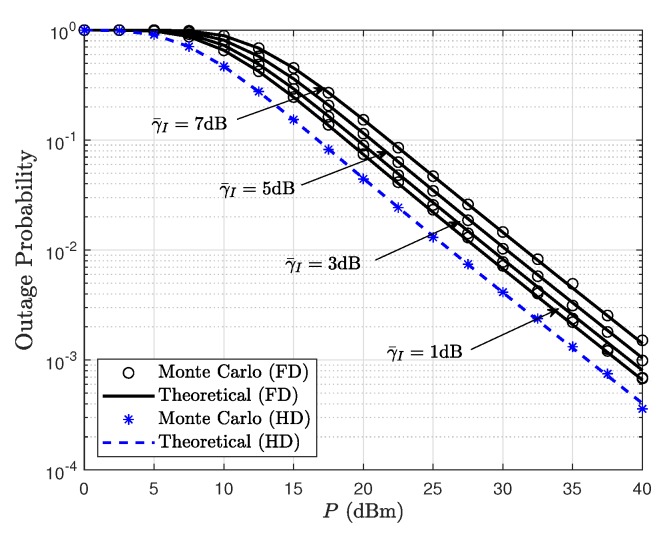
Outage probability comparison between FD and HD satellite relaying systems for different values of ${\bar{\gamma }}_{I}$ with $\theta_{1}=\theta_{2}=20^{\circ }$, $\varphi_{1}=\varphi_{2}=0.4^{\circ }$, and $\gamma_{th}=0$ dB.

**Figure 3 sensors-19-05453-f003:**
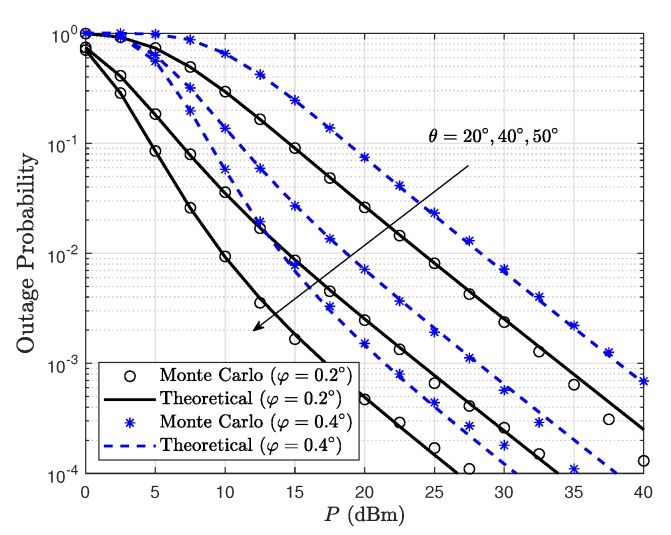
Outage probability of FD satellite relaying system for different $\varphi_{}$ and $\theta_{}$ with ${\bar{\gamma }}_{I}=1$ dB and $\gamma_{th}=0$ dB.

**Figure 4 sensors-19-05453-f004:**
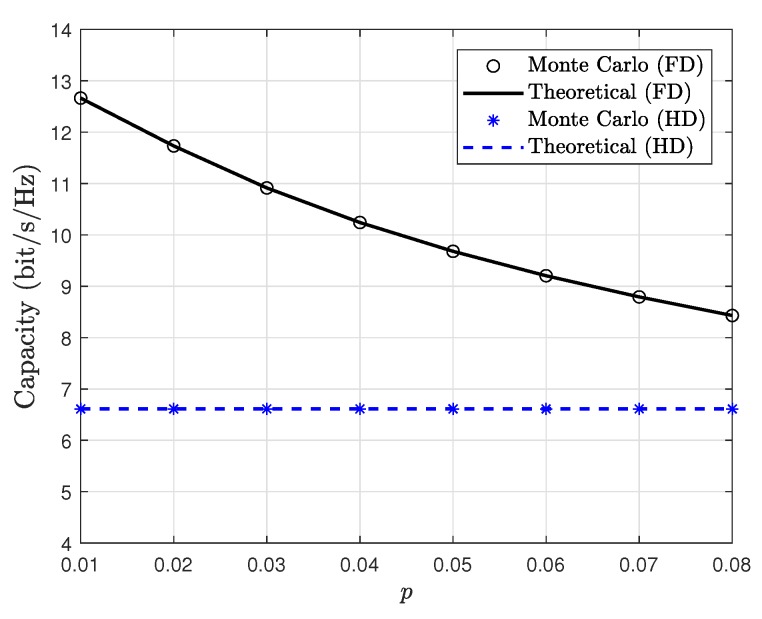
Ergodic capacity comparison between FD and HD satellite relaying system versus *p* with $\theta_{1}=\theta_{2}=20^{\circ }$, $\varphi_{1}=\varphi_{2}=0.2^{\circ }$.

**Figure 5 sensors-19-05453-f005:**
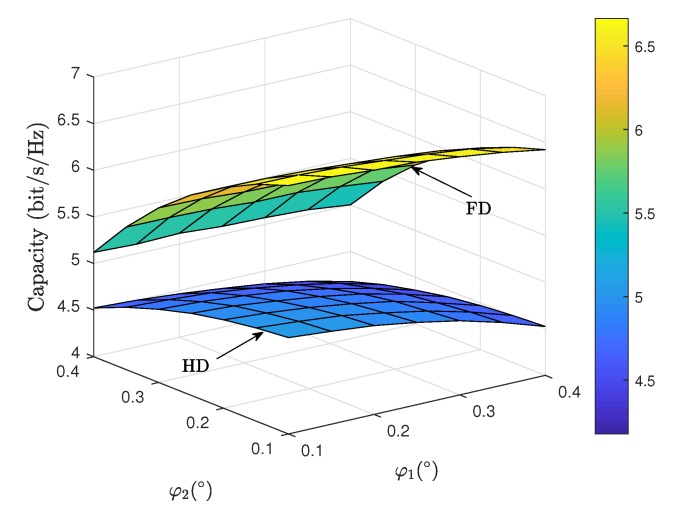
Ergodic capacity comparison between FD and HD satellite relaying systems versus $\varphi_{1}$ and $\varphi_{2}$ with p=5%.

**Figure 6 sensors-19-05453-f006:**
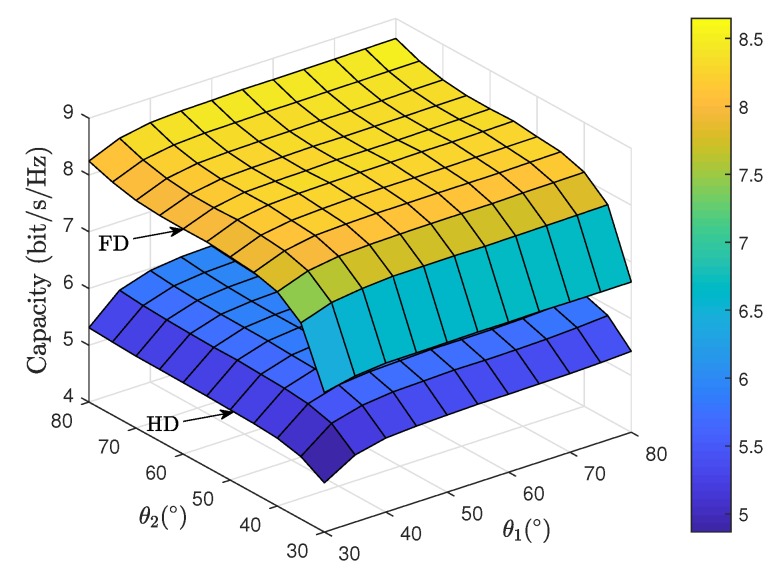
Ergodic capacity comparison between FD and HD satellite relaying systems versus $\theta_{1}$ and $\theta_{2}$ with p=5%.

**Table 1 sensors-19-05453-t001:** Echo residual interference.

Error Probability ε	Cancellation Performance
0.5%	46 dB
1%	40 dB
2%	34 dB

**Table 2 sensors-19-05453-t002:** System Parameter [33,34,35].

Frequency Band $f_{c}$	2 GHz
Total downlink Bandwidth *B*	1 MHz
Satellite 3-dB angel $\varphi_{3\mathrm{dB}}$	$0.3^{\circ }$
Maximum satellite beam gain	52.1 dBi
ES antenna gain	42.1 dBi
Base station noise temperature	300 K
Satellite noise temperature	350 K

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
