# Peer review of "Performance Evaluation of a Full-Duplex Relaying-Enabled Satellite Sensor Network"

_sensors, 2019, doi:10.3390/s19245453_

Round 1

Reviewer 1 Report

This paper investigates the performance of a full-duplex (FD) relaying enabled satellite sensor network under residual loop interference, where the satellite uplink and the downlink transmissions simultaneously occur over the same frequency band.

In general, the paper is well written and easy to follow.

The analysis seems solid and are verified by simulation.

The systems’ parameters are not well defined and justified.

First, the authors must mention clearly that it is a geo satellite otherwise Doppler effect must be included in the simulation

Second, the authors did not consider any nonlinear component coming for the TWTA. This impact can be just shown by simulation

The uplink and downlink frequencies must be different please update your model.

Why the atmosphere and weather parameters are omitted from the channel model. Please justify

Normally satellite gains are a bit higher than 52 dB.

Why the temperature is only 300K. In fact, the temperature at the satellite is higher than the temperature at base station nodes, please justify.

Finally, the literature review needs an update as similar performance analysis papers have been published deserve to be discussed.  

Fidan et al, "Performance of Transceiver Antenna Selection in Two Way Full-Duplex Relay Networks Over Rayleigh Fading Channels," in IEEE Transactions on Vehicular Technology, vol. 67, no. 7, pp. 5909-5921, July 2018.

Eshteiwi, et al, "Performance Analysis of Full-Duplex Vehicle Relay-Based Selection in Dense Multi-Lane Highways," in IEEE Access, vol. 7, pp. 61581-61595, 2019

Eshteiwi, et al, "Performance analysis of peer-to-peer V2V wireless communications in the presence of interference," 2017 IEEE 28th Annual International Symposium on Personal, Indoor, and Mobile Radio Communications (PIMRC), Montreal, QC, 2017, pp. 1-6.

Author Response

Dear reviewer, 

We really appreciate the time and effort devoted, whichi helps us to improve the quality of our paper significantly. We have accordingly addressed all comments raised. Please find our detailed point-by-point response in the attachment.

Thank you and with our best regards

Co-authors

Reviewer 2 Report

This paper considers a residual loop interference problem between base stations in a satellite sensor networks. They analyzed a performance analysis of full-duplex communications based on the transmission power of base stations and satellites. There are several shortcomings in this paper.

Basically in satellite networks, most base stations use directional antennas to communicate with satellites. Because the antennas have an ultra narrow beam-width, the residual loop interference between terrestrial base stations is a negligible quantity. So, the fundamental assumption of the paper is not appropriate. There is no definition of UT and GW on page 3. There is a lack of explanation of how equation (3) is derived. The derivation process and solution direction of many equations are somewhat difficult to understand. Authors need to double check the format and organization of this paper.

Author Response

(The authors gave the same response as above.)

Round 2

Reviewer 1 Report

In my opignion, the paper can be considered for publication

Author Response

(The authors gave the same response as above.)

Reviewer 2 Report

Most on-board satellite antennas use directional antennas. So, the self-interference between the on-board satellite antennas for transmitting and receiving is less affected. If there is enough self-interference between the on-board satellite antennas, this study is not different from the existing full-duplex relay network papers. It is necessary to present the differences from the existing papers for the performance analysis in full-duplex wireless relay networks. The authors should check overall about the paper format of this journal (Sensors).

Author Response

(The authors gave the same response as above.)

Round 3

Reviewer 2 Report

The paper can be accepted in this version.